# Gut Microbiota and Subclinical Cardiovascular Disease in Patients with Type 2 Diabetes Mellitus

**DOI:** 10.3390/nu13082679

**Published:** 2021-08-01

**Authors:** Hui-Ju Tsai, Wei-Chung Tsai, Wei-Chun Hung, Wei-Wen Hung, Chen-Chia Chang, Chia-Yen Dai, Yi-Chun Tsai

**Affiliations:** 1Department of Family Medicine, Kaohsiung Municipal Ta-Tung Hospital, Kaohsiung Medical University Hospital, Kaohsiung Medical University, Kaohsiung 801, Taiwan; bankin_0920@yahoo.com.tw; 2Department of Family Medicine, School of Medicine, College of Medicine, Kaohsiung Medical University, Kaohsiung 807, Taiwan; 3Research Center for Environmental Medicine, Kaohsiung Medical University, Kaohsiung 807, Taiwan; 4Division of Cardiology, Department of Internal Medicine, Kaohsiung Medical University Hospital, Kaohsiung Medical University, Kaohsiung 807, Taiwan; azygo91@gmail.com; 5Department of Microbiology and Immunology, College of Medicine, Kaohsiung Medical University, Kaohsiung 807, Taiwan; wchung@kmu.edu.tw (W.-C.H.); hung4488@ms57.hinet.net (C.-C.C.); 6Division of Endocrinology and Metabolism, Department of Internal Medicine, Kaohsiung Medical University Hospital, Kaohsiung Medical University, Kaohsiung 807, Taiwan; wacsun@gmail.com; 7School of Medicine, College of Medicine, Kaohsiung Medical University, Kaohsiung 807, Taiwan; daichiayen@gmail.com; 8Division of Hepatobiliary, Department of Internal Medicine, Kaohsiung Medical University Hospital, Kaohsiung Medical University, Kaohsiung 807, Taiwan; 9Division of General Medicine, Department of Internal Medicine, Kaohsiung Medical University Hospital, Kaohsiung Medical University, Kaohsiung 807, Taiwan; 10Liquid Biopsy and Cohort Research Center, Kaohsiung Medical University, Kaohsiung 807, Taiwan

**Keywords:** type 2 diabetes mellitus, gut microbiota, subclinical cardiovascular disease

## Abstract

Type 2 diabetes (T2D) is associated with an increased risk of cardiovascular disease (CVD). The gut microbiota may contribute to the onset and progression of T2D and CVD. The aim of this study was to evaluate the relationship between the gut microbiota and subclinical CVD in T2D patients. This cross-sectional study used echocardiographic data to evaluate the cardiac structure and function in T2D patients. We used a quantitative polymerase chain reaction to measure the abundances of targeted fecal bacterial species that have been associated with T2D, including *Bacteroidetes, Firmicutes, Clostridium leptum group, Faecalibacterium prausnitzii, Bacteroides, Bifidobacterium, Akkermansia muciniphila,* and *Escherichia coli.* A total of 155 subjects were enrolled (mean age 62.9 ± 10.1 years; 57.4% male and 42.6% female). Phyla *Bacteroidetes* and *Firmicutes* and genera *Bacteroides* were positively correlated with the left ventricular ejection fraction. Low levels of phylum *Firmicutes* were associated with an increased risk of left ventricular hypertrophy. High levels of both phylum *Bacteroidetes* and genera *Bacteroides* were negatively associated with diastolic dysfunction. A high phylum Firmicutes/*Bacteroidetes* (F/B) ratio and low level of genera *Bacteroides* were correlated with an increased left atrial diameter. Phyla *Firmicutes* and *Bacteroidetes*, the F/B ratio, and the genera *Bacteroides* were associated with variations in the cardiac structure and systolic and diastolic dysfunction in T2D patients. These findings suggest that changes in the gut microbiome may be the potential marker of the development of subclinical CVD in T2D patients.

## 1. Introduction

Type 2 diabetes (T2D) and its complications are major global health concerns. The global prevalence of T2D has increased by 62% in the last 10 years, and it is predicted to reach 10.9% or 700 million people among those aged 20–79 years by 2045 [1]. T2D increases the risk of cardiovascular disease (CVD) and heart failure (HF), which are the major causes of morbidity and mortality in patients with T2D [2,3,4]. In addition, accumulating evidence has suggested an association between T2D and an increased risk of subclinical CVD, including vascular calcification, heart dysfunction and an abnormal heart structure [2].

The human gut microbiota consists of trillions of bacteria residing in the gastrointestinal tract. It is regarded to be a microbial organ with functions including immune regulation, material absorption, and energy metabolism, and it has been shown to play an important role in human health and disease [5,6]. Dysbiosis has been associated with many diseases, including T2D and CVD, by provoking changes in the composition of the gut microbiota, altering the intestinal wall permeability and enhancing the secretion of metabolic endotoxins [7,8,9]. Phyla *Bacteroidetes Firmicutes* account for more than 90% of the whole gut microbiota [7,10]. However, the proportion of the phylum *Firmicutes* is significantly lower in T2D patients compared to healthy individuals, and the phyla *Firmicutes*-to-*Bacteroidetes* ratio (F/B ratio) has been significantly associated with impaired glucose metabolism [11]. Lower abundances of *Faecalibacterium prausnitzii* and *Bifidobacterium* species have also been reported in T2D patients. Both *Faecalibacterium prausnitzii* and *Bifidobacterium* species have been linked to butyrate production, and butyrate production has been associated with a local anti-inflammatory effect in the intestinal mucosa [12,13,14,15]. T2D patients have been reported to have a higher abundance of *Escherichia coli* [13]. In addition, *Akkermansia muciniphila* may contribute to the treatment of T2D through glucose tolerance and adipose tissue inflammation [16].

Previous studies have suggested that alterations in the gut microbial community may play a role in CVD [17,18]. Dysbiosis causes inflammation and metabolic disorders, further promoting the development of CVD [18,19,20]. For example, Cui et al. reported a high phyla F/B ratio in patients with heart disease [21]. In addition, patients with HF had relative depletion of phylum *Bacteroidetes* and genera *Bacteroides* [17]. Moreover, a lower abundance of *Faecalibacterium prausnitzii* and higher abundances of the *Enterobacteriaceae* family have been reported in patients with CVD and HF [22,23,24]. Furthermore, Kummen et al. also reported a lower abundance of *Bifidobacterium* in patients with HF [25]. Taken together, these findings show that imbalance of the gut microbiota is closely correlated with clinical CVD. However, the impact of the gut microbiota on subclinical CVD is unclear in T2D patients.

Since T2D can lead to cardiac structural changes before the development of clinical CVD, and the gut microbiota may play a role in CVD, the aim of this study was to evaluate the relationship between the gut microbiota and subclinical CVD in T2D patients.

## 2. Materials and Methods

### 2.1. Study Participants

We invited patients with T2D who attended the outpatient clinics of a tertiary hospital in Southern Taiwan from October 2016 to August 2017 to participate in this observational study. A diagnosis of T2D was based on a previous history of T2D, blood glucose values defined by the American Diabetes Association, or the use of antidiabetic drugs, as in our previous study [26]. All of the patients were enrolled in an education program on diabetes at our hospital, and they followed the guidelines for a diabetic diet. We excluded patients who used antibiotics within 1 month prior to enrollment, the same as previous studies [27,28,29]. In addition, patients with inflammatory bowel disease, those who had received gastrointestinal tract surgery, and those with cancer who had undergone chemotherapy within 1 year prior to enrollment were also excluded. A total of 155 T2D patients were included in this study. The study protocol was approved by the Institutional Review Board of Kaohsiung Medical University Hospital (KMUHIRB-G(II)-20160021). All of the included patients provided written informed consent to participate in the study, and all clinical investigations were conducted according to the principles expressed in the Declaration of Helsinki.

### 2.2. Clinical Measurements

The following information was obtained from medical records and interviews with the patients at enrollment: age, sex, the use of medications, tobacco and alcohol use, and comorbidities. Heart disease was defined as a history of congestive HF, ischemic heart disease, or myocardial infarction. Hypertension was defined as a previous history of hypertension, blood pressure ≥140/90 mmHg, or the use of antihypertensive drugs. Hyperlipidemia was defined as a previous history of hyperlipidemia, triglyceride level ≥160 mg/dL, cholesterol level ≥200 mg/dL, or the use of anti-hyperlipidemia drugs. Data on medications, including antidiabetic agents, statins, angiotensin II receptor blockers (ARB), angiotensin-converting enzyme inhibitors (ACEI), β-blockers, and calcium channel blockers before and after enrollment, were obtained from the medical records. Blood pressure was measured after the patients had been allowed to rest 5 min and when seated. A single calibrated device was used, and the mean of three consecutive blood pressure measurements made at 5-min intervals was used for the analysis. The body mass index (BMI) was calculated as the body weight/body height squared (kg/m^2^). Information on the usual diet of the patients was obtained using a simple questionnaire. Blood and urine samples were taken after a 12-h fast for biochemistry studies on the same day as stool collection. The urine albumin levels were measured using the urine albumin–creatinine ratio (ACR).

### 2.3. Collection of Stool Samples and Extraction of Microbial DNA

The patients were asked to collect fecal samples at home, immediately place them in their household freezer, and then bring them to the hospital within 12 h. All fecal samples were collected either on the evening before or the morning of the biochemical tests. After the patients brought the samples to the hospital, they were stored at −80 °C for up to 3 days before being processed. A Stool DNA Extraction kit (Topgen Biotechnology Co., Ltd., Kaohsiung, Taiwan) was used to extract bacterial DNA. In brief, 50–100 mg of the fecal samples were subjected to bead-beating (45 s; speed: 3450 oscillations/min). DNA was then extracted according to the manufacturer’s instructions. The purified DNA was eluted in a volume of 35 μL. The concentration and quality of the DNA were evaluated using a Colibri Microvolume spectrophotometer (Titertek Berthold, Pforzheim, Germany). The DNA samples were then immediately frozen at −20 °C until use.

### 2.4. Real-Time Quantitative Polymerase Chain Reaction (qPCR)

Bacterial 16S rRNA genes in the fecal samples were quantified using real-time qPCR on a StepOnePlus Real-Time PCR system (Thermo Fisher Scientific, Waltham, MA, USA), as previously reported [26,30,31,32]. Appendix A lists the eight pairs of 16S rRNA gene primers specific to the *C. leptum* group: *E. coli*, *Bacteroidetes*, *Firmicutes*, *Bacteroides*, *F. prausnitzii, Bifidobacterium*, and *A. muciniphila* that were used in this study. Standard curves were generated using 10-fold dilutions of the 16S rRNA gene fragments amplified from the reference strains that were cloned into a T&A^TM^ Cloning Vector (Yeastern Biotech, Co., Ltd., Taipei, Taiwan). Each reaction mixture had a total volume of 10 μL and contained 1 μL of sample DNA, 5 μL of AceQ qPCR SYBR Green Master Mix (Vazyme Biotech Co., Piscataway, NJ, USA), 0.25 μL of each 10-μM primer, and 3.5 μL of sterilized ultra-pure water. The cycle conditions of the real-time PCR were as follows: initial holding at 95 °C for 30 s, 40 cycles of denaturation at 95 °C for 3 s, followed by annealing/elongation at 60 °C for 40 s. The specificity was determined after amplification by a melting curve analysis. Quantitation of the eight target bacteria was determined as copy numbers of 16S rDNA per gram of feces. All qPCR tests were performed in duplicate, and the mean values were used for analysis.

### 2.5. Measurement of Cardiac Structure and Function

The cardiac structure and function were evaluated as previously reported [33]. In brief, the patients underwent echocardiographic examinations on the same day as they provided urine and blood samples. All of the examinations were performed by experienced cardiologists who were blind to the clinical characteristics and laboratory data of the patients using a VIVID 7 ultrasound system (General Electric Medical Systems, Horten, Norway), with the patients breathing quietly in the left decubitus position. M-mode and two-dimensional images were obtained from the standard views. The following echocardiographic parameters were obtained: left ventricular internal diameter in systole and diastole, left ventricular posterior wall thickness in systole and diastole, left atrial diameter (LAD), interventricular septal wall thickness in systole and diastole, peak early transmitral filling wave velocity (E), and peak late transmitral filling wave velocity (A). Left ventricular systolic function was assessed using left ventricular ejection fraction (LVEF) and left ventricular fractional shortening (LVFS), and the Devereux-modified method was used to calculate left ventricular mass (LVM) [34], The left ventricular mass index (LVMI) was calculated as LVM divided by body surface area. Left ventricular hypertrophy (LVH) was defined according to the 2007 European Society of Hypertension/European Society of Cardiology guidelines [35]. Stroke volume was calculated as the volume of blood just prior to the beat (end-diastolic volume) minus the volume of blood in the ventricle at the end of a beat (end-systolic volume). Diastolic dysfunction was defined as an E/A ratio <1. As early mitral inflow velocity (E) and mitral annular early diastolic velocity (E’) can easily be measured; the data were obtained from five beats and then averaged for analysis [36].

### 2.6. Statistical Analysis

Continuous variables are presented as the mean ± SD or median (25th and 75th percentiles), and those with skewed distribution were log-transformed to achieve normal distribution. Categorical variables are presented as percentages. We performed a principal components analysis (PCA) to examine the relationship between the target microbiota and echocardiographic parameters, while missing values were imputed with a regularized expectation–maximization algorithm via the function imputePCA in the R package missMDA [37]. Associations between the target microbiota and echocardiographic parameters were examined using linear and logistic regression analyses with three stepwise models. The first model included age and sex. Model 1 included age, sex, tobacco and alcohol use, BMI, hypertension, the use of β-blocker, ACRI/ARB, and statin. Model 2 was adjusted for the variables of model 1 plus high-density lipoprotein (HDL), log-formed low-density lipoprotein (LDL), HbA1C, log-formed urine albumin-creatinine ratio (UACR), and log-formed triglyceride (TG). We used forest plots to present the results of the multivariate regression. Statistical analyses were conducted using SPSS for Windows version 18.0 (SPSS Inc., Chicago, IL, USA). Statistical significance was set at a two-sided *p*-value of <0.05.

## 3. Results

### 3.1. Characteristics of the Entire Cohort

Table 1 shows the clinical characteristics, medication records, and laboratory parameters of the entire cohort. Of the 155 enrolled subjects, the mean age was 62.9 ± 10.1 years, 57.4% were male, 42.6 % were female, and the median of duration of diabetes was 10.0 (5.0–15.0) years. The prevalence rates of heart disease, hypertension and hyperlipidemia were 20.6%, 67.1% and 82.6%, respectively. The median glycated hemoglobin level was 7.0%. The medium bacterial concentrations (estimated as copy numbers of 16S rDNA per gram of feces) are also listed in Table 1.

Table 2 shows the distribution of the echocardiographic parameters. The median LVMI was 95.8 g/m^2^, and 18.7% of the study cohort had LVH. The mean LVEF and LVFS values were 70.6% ± 10.1% and 40.9% ± 8.1%, respectively, and 79.9% of the study cohort had an E/A ratio <1. The mean E/E’ ratio was 9.2 ± 2.8.

### 3.2. Correlations between Targeted Microbiota and LV Structure and LV Systolic Function

We examined the relationship between the echocardiographic parameters and eight targeted bacteria in univariate correlation analysis. Figure 1 showed phylum *Firmicutes* was negatively associated with LVMI, and phylum *Bacteroidetes*, genera *Bacteroides* and F/B ration were negatively related to LA diameter using PCA.

The univariable-adjusted unstandardized β-values of LVEF and LVFS were significant for every increase in the log-transformed phylum *Bacteroidetes* (β = 3.80, *p* = 0.02; β = 2.88, *p* = 0.04, respectively; Appendix A). The phylum *Bacteroidetes* was positively and significantly correlated with the LVEF (β = 3.92, *p* = 0.03) and LVFS (β = 2.99, *p* = 0.04) after adjusting the age, sex, tobacco and alcohol use, BMI, hypertension, the use of a β-blocker, ACRI/ARB, statin, HDL, log-formed LDL, HbA1C, UACR, and log-formed TG in the multivariate linear regression model 2 (Figure 2 and Appendix A). Positive associations of the LVEF with the phylum *Firmicutes* (β = 4.58, *p* = 0.03) and genera *Bacteroides* (β = 3.36, *p* = 0.04) were shown in the multivariate-adjusted model 1, including the age, sex, tobacco and alcohol use, BMI, hypertension, the use of a β-blocker, ACRI/ARB, and statin but not in model 2 (Appendix A). Although there were no significant correlations between the LVMI and the eight targeted bacteria, the high levels of the phylum *Firmicutes* (odds ratio (OR) (95% confidence index (CI)): 0.24 (0.06–0.95), *p* = 0.04) were associated with a reduced risk of LVH after adjustment (Figure 2 and Appendix A). Figure 2 showed a significant association between the phylum *Firmicutes*, phylum *Bacteroidetes*, F/B ratio, genera *Bacteroides*, LV structure, and LV systolic function after adjustment. There were no significant associations of the LV structure and systolic function with the *C. leptum* group, genera *Bifidobacterium*, *F. prausnitzii*, *A muciniphila*, or *E coli*.

### 3.3. Correlations between the Targeted Bacteria and Diastolic Function

The phylum *Bacteroidetes* (β = −1.30, *p* = 0.02) and genera *Bacteroides* (β = −1.06, *p* = 0.03) were negatively and significantly correlated with the E/E’ (Figure 3 and Appendix A). According to the low percentage of LV diastolic dysfunction, defined as E/E’ > 15 [38], we stratified the patients by the median E/E’ (8.84) and found that high levels of the phylum *Bacteroidetes* (OR (95% CI): 0.33 (0.14–0.81), *p* = 0.02) and genera *Bacteroides* (OR (95% CI): 0.42 (0.19–0.93), *p* = 0.03) were associated with a lower risk of E/E’ > median in the multivariable-adjusted logistic regression. However, neither was correlated with E/A < 1, which is an indicator of diastolic dysfunction. In addition, there were no significant associations of diastolic dysfunction with the *C. leptum* group, genera *Bifidobacterium*, *F. prausnitzii*, *A muciniphila*, or *E coli*.

### 3.4. Correlations between the Targeted Bacteria and Left Atrium Structure

The univariate and multivariable-adjusted unstandardized β-values of the LAD were also significant for every decrease in the log-transformed genera *Bacteroides* (β = −0.29, *p* = 0.006; β = −0.27, *p* = 0.008, respectively (Figure 4 and Appendix A). We further stratified the patients by the median LAD (3.74 cm) and found that a high phyla F/B ratio (OR (95% CI): 3.10 (1.08–8.95), *p* = 0.04) and low abundance of genera *Bacteroides* (OR (95% CI): 0.30 (0.13–0.70), *p* = 0.006) were correlated with an increased risk of LAD > median in multivariable-adjusted logistic regression. There were no significant associations of the LA structure with the *C. leptum* group, genera *Bifidobacterium*, *F. prausnitzii*, *A muciniphila*, or *E coli*.

## 4. Discussion

T2D patients tend to have diastolic dysfunction, subclinical systolic dysfunction, and LVH [39]. This study is the first to explore the associations between the gut microbiota and subclinical features of CVD in T2D patients. A low abundance of phylum *Bacteroidetes* was significantly correlated with LV systolic and diastolic dysfunction. In addition, low abundances of phylum *Firmicutes* were associated with a higher risk of LVH. We also found that a high phyla F/B ratio and low abundance of the genera *Bacteroides* were significantly correlated with LA enlargement. These findings suggest that alterations in the gut microbial composition may be associated with the echocardiographic findings of subclinical CVD and, also, that the gut microbiota might be used to predict the development of subclinical CVD in T2D patients.

Previous studies have reported associations between alterations in the gut microbial composition and HF and CVD [17]. Yuzefpolskaya et al. reported that patients with HF had a relative depletion of the phylum *Bacteroidetes* [40]. Kamo et al. also reported that the phylum *Bacteroidetes* was less abundant in the gut microbiota of older HF patients than in that of younger patients and that there was no significant difference in the phylum *Firmicutes* between the two groups [41]. In addition, Yoshida et al. reported that patients with coronary artery disease or HF had a lower abundance of genera *Bacteroides* [42]. The genera *Bacteroides* has been shown to be present in atherosclerotic plaques, and it has been suggested to be a diagnostic marker in patients with coronary artery disease [43]. However, few studies have explored the relationship between the gut microbiota and cardiac structure and function in asymptomatic patients with a high-risk of CVD, such as those with T2D. In the present study, we found that the patients with a low abundance of phylum *Bacteroidetes* had LV systolic and diastolic dysfunction. In addition, our results indicated that low abundances of the phylum *Firmicutes* were associated with a higher risk of LVH, and a low abundance of genera *Bacteroides* was correlated with LA enlargement. Thus, alterations in the gut microbiota components may affect the cardiac structure and function in asymptomatic T2D patients, suggesting that the gut microbiota may be a potential biomarker to detect subclinical CVD in high-risk patients.

Our results indicated the relationship between subclinical CVD and low abundances of the phyla *Firmicutes* and *Bacteroidetes* and genera *Bacteroide*s in patients with T2D after adjusting for traditional risk factors of CVD. A previous study reported correlations between acetate, propionate, and butyrate (major short-chain fatty acids (SCFAs) with CVD [44]. The main butyrate-producing bacteria in the human gut belong to the phylum *Firmicutes*, while the phylum *Bacteroidetes* mainly produces acetate and propionate, and the genera *Bacteroides* mainly produces acetate. Both butyrate and propionate can help regulate the vascular tone and blood pressure through receptors on smooth muscle cells [45]. Butyrate has an anti-inflammatory effect and maintains an intestinal barrier integrity by modulating the function of intestinal macrophages and downregulating lipopolysaccharide (LPS)-induced proinflammatory mediators [46]. Butyrate can also induce the differentiation of regulatory T cells, which has been shown to suppress the progression of HF [47,48]. On the other hand, propionate has been shown to mitigate pressure-induced cardiac hypertrophy and fibrosis by maintaining immune homeostasis through regulatory T-cell activation [49]. Acetate has been shown to exert cardioprotective effects by regulating the genes involved in cardiac fibrosis and cardiac hypertrophy [50]. Therefore, reductions in SCFA-producing bacteria, including the phyla *Firmicutes* and *Bacteroidetes* and the genera *Bacteroide*s, may aggravate the inflammatory status and be associated with the development of an abnormal cardiac structure and cardiac dysfunction in T2D patients.

In the current study, asymptomatic T2D patients with a low abundance of the genera *Bacteroides* had a high risk of an abnormal left ventricle structure, including LA enlargement and impaired diastolic function. Accumulating evidence has demonstrated a link between the gut microbiota and HF, mainly through intestinal barrier impairment and bacterial translocation to induce inflammation and the immune responses [51]. Intestinal epithelial cells can be damaged by intestinal ischemia, and epithelial dysfunction further impairs the absorption of sugar, protein, and fat, which may have an adverse effect on the development of HF [52]. In addition, an increased potential for SCFA biosynthesis in the microbiome and elevated serum LPS levels have been reported among patients with CVD and HF [24]. Jie et al. reported an increase in the genes required for LPS O-antigen synthesis and a depleted lipid A module in patients with CVD and HF. The possible mechanism underlying the relationship between the genera *Bacteroides* and subclinical CVD may be through the depletion of the genera *Bacteroides* and the production of noninflammatory penta-acylated lipid A [22].

The F/B ratio has been reported to be a potential marker of pathophysiologic conditions [17,53]. Mayerhofer et al. suggested that patients with HF have a lower F/B ratio than normal individuals [54]. In the present study, a higher F/B ratio was associated with LA enlargement in asymptomatic T2D patients, although we did not find a relationship between the F/B ratio and LV function and structure. Previous studies have demonstrated the depletion of *F. prausnitzii* in HF patients [23,25]. *F. prausnitzii* is one of the most abundant butyrate-producing species, and a lack of *F. prausnitzii* may aggravate chronic inflammation. However, we did not find a significant correlation between a reduction in *F. prausnitzii* and LV dysfunction in our asymptomatic T2D patients. This inconsistency may be related to different severities of CVD and different populations. We also explored the associations of cardiac structure and function with the abundances of other T2D-related gut microbiota, including *Bifidobacterium*, *A. muciniphila*, and *E. coli*, but we did not find any significant associations. These findings suggest that the interactions between alterations in the gut microbiota components and subclinical CVD differ in T2D patients. In addition, the relationships among the gut microbiota, T2D, and CVD are complex. Insulin resistance and hyperglycemia induce endothelial dysfunction and a proinflammatory status, thereby contributing to cardiovascular dysfunction [54]. Thus, interactional effects between the gut microbiota and T2D may affect the development of subclinical CVD and progression to CVD [17]. Further investigations are necessary to confirm these relationships.

The correlation between the gut microbiota and subclinical CVD is complicated and might be affected by many factors, such as diet habits and medications. For example, previous studies suggested that metformin may be related to a higher abundance of *A muciniphila* and several butyrate-producing bacteria to maintain the glycemic status in T2D patients [55,56]. Kim et al. reported that statin therapy may increase the abundance of the genera *Bacteroides* [57]. This study collected information on diet habits and medications. All of the patients were enrolled in an education program on diabetes at our hospital, and they followed the guidelines for a diabetic diet to minimize the effect of dieting on the gut microbiota. We analyzed the correlation between medication and the target bacteria in these patients and did not find a significant effect of medication on the target bacteria concentration. In addition, accumulating evidence has indicated an impact of the β-blocker, ACRI/ARB, and statin on subclinical CVD [58]. Hence, we adjusted the use of the β-blocker, ACRI/ARB, and statin in the multivariate regression model and found that the imbalance composition of the gut microbiota was associated with an abnormal cardiac structure and function.

There are several limitations to this study. First, the gut microbiota and echocardiographic parameters were measured only once at enrollment. Therefore, the time-varying relationship between the gut microbiota and cardiac structure and function could not be evaluated. Second, the relatively small number of T2D subjects and a single hospital for recruitment may have underestimated the association between the gut microbiota and subclinical CVD. In addition, this study lacked the composition of the gut microbiota of healthy individuals matched by age and gender to compare with those of the T2D patients. Thirdly, several confounding factors affected the abundance of the microbiota. Although we adjusted several factors, including the age, sex, tobacco and alcohol use, BMI, hypertension, medication, and lab data related to CVD, some confounders may not be considered. Finally, this study only measured eight targeted gut bacteria using real-time qPCR. There may be other bacteria related to echocardiographic changes that we did not examine. Lacking a next-generation sequencing (NGS) analysis of the gut microbiota is one of our limitations. NGS is a hypothesis-free method that does not require prior knowledge of sequence information and provides a higher discovery power to detect novel genes and higher sensitivity to quantify rare variants and transcripts. A NGS analysis might help us to realize all picture of the relative composition of the gut microbiota. Since the aim of our study was to analyze the relationship between the target bacteria and subclinical CVD, a qPCR is typically a good choice when the number of target regions is low (≤20 targets) and when the study aims are limited to identifying the known variants. Nevertheless, further study is necessary to perform a NGS analysis to evaluate the compositional differences of the gut microbiota between T2D patients and healthy individuals matched by age and gender and show the alpha, beta, and overall compositional changes to provide more detailed information of the correlations between the echocardiographic changes and gut microbiota.

## 5. Conclusions

In conclusion, in this study, we demonstrated that phyla *Firmicutes* and *Bacteroidetes*, F/B ratio, and genera *Bacteroides* were correlated with echocardiographic changes in subclinical CVD, including systolic and diastolic dysfunction and left atrium and ventricular hypertrophy, in asymptomatic T2D patients. The imbalance of gut microbiota might be a potential biomarker of the development of subclinical CVD in patients with T2D.

## Figures and Tables

**Figure 1 nutrients-13-02679-f001:**
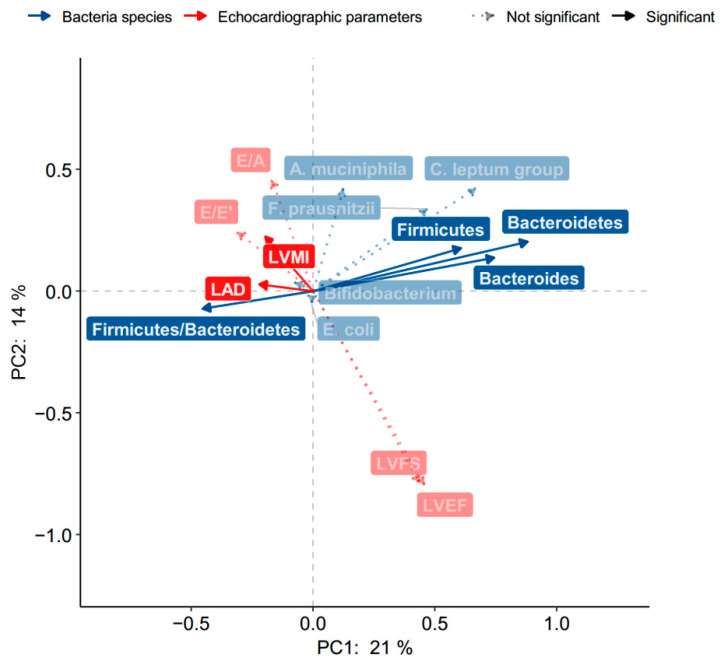
The correlations between the echocardiographic parameters and eight targeted bacteria in the principal component analysis. Dark blue lines represent bacteria species, and dark red lines indicate echocardiographic parameters; the significance of the correlation coefficients is denoted by the transparency and line type.

**Figure 2 nutrients-13-02679-f002:**
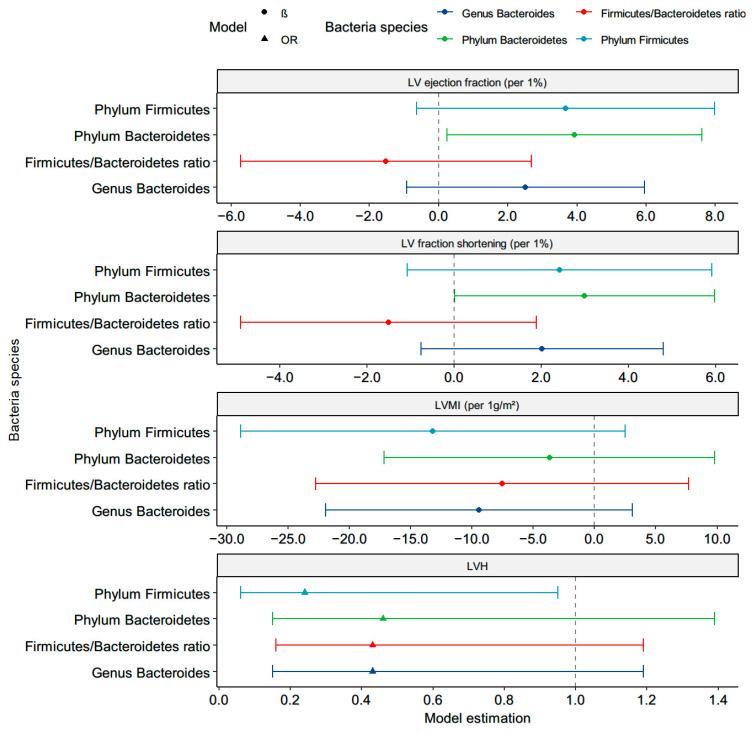
Correlation between the targeted microbiota and left ventricular (LV) structure and function in the adjusted model. β or OR were adjusted for age, sex, smoke, alcohol, body mass index, hypertension, β-blocker, ACRI/ARB, statin, high-density lipoprotein, log-formed low-density lipoprotein, HbA1C, log-formed urine albumin-creatinine ratio, and log-formed triglyceride.

**Figure 3 nutrients-13-02679-f003:**
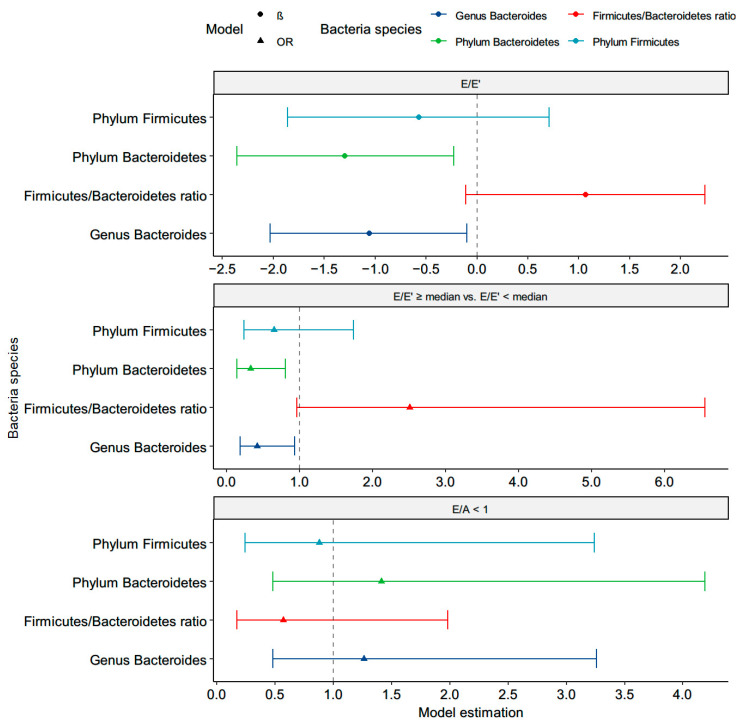
Correlation between the targeted microbiota and left ventricular diastolic function in the adjusted model. β or OR were adjusted for age, sex, smoke, alcohol, body mass index, hypertension, β-blocker, ACRI/ARB, statin, high-density lipoprotein, log-formed low-density lipoprotein, HbA1C, log-formed urine albumin-creatinine ratio, and log-formed triglyceride.

**Figure 4 nutrients-13-02679-f004:**
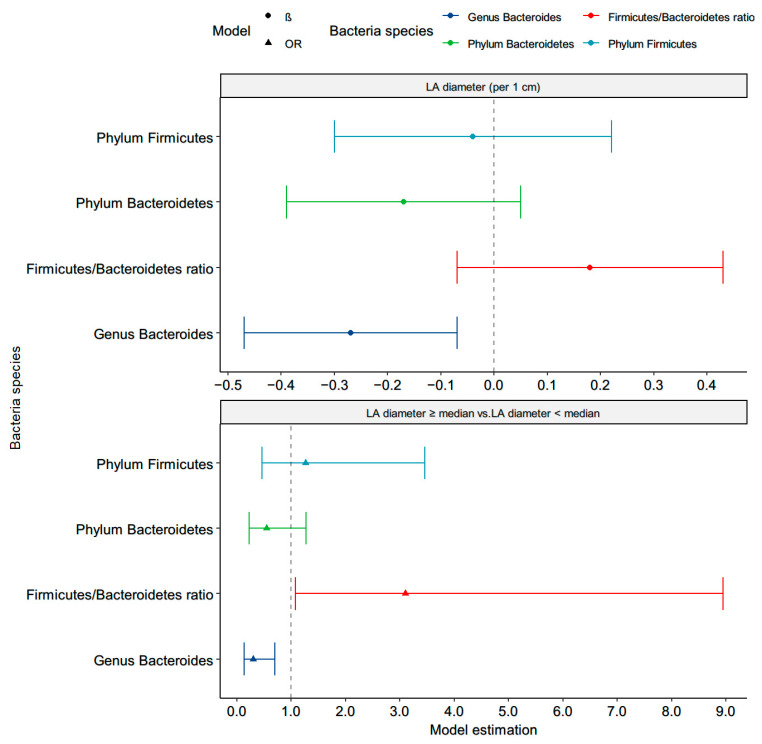
Correlation between the targeted microbiota and left atrium diameter in the adjusted model. β or OR were adjusted for age, sex, smoke, alcohol, body mass index, hypertension, β-blocker, ACRI/ARB, statin, high-density lipoprotein, log-formed low-density lipoprotein, HbA1C, log-formed urine albumin-creatinine ratio, and log-formed triglyceride.

**Table 1 nutrients-13-02679-t001:** The clinical characteristics of the study subjects.

Clinical Characteristics	Entire Cohort N = 155
Age, year	62.9 ± 10.1
Sex, %	
female	42.6
male	57.4
Smoke, %	20.9
Alcohol, %	16.3
Heart disease, %	20.6
Hypertension, %	67.1
Hyperlipidemia, %	82.6
Body mass index, kg/m^2^	26.2 ± 4.0
Systolic blood pressure, mmHg	137.7 ± 17.4
Diastolic blood pressure, mmHg	79.1 ± 10.3
Diabetic duration, years	10.0 (5.0,15.0)
Diet habit, %	
Protein more than fiber	14.3
Fiber more than protein	37.5
Fiber equal to protein	48.2
Medications	
Sulfonylurea user, %	52.3
Metformin user, %	86.5
Dipeptidyl peptidase-4-inhibitor user, %	65.2
Thiazolidinediones, %	58.7
Insulin user, %	11.6
Calcium channel blocker user, %	20.6
Beta-blocker user, %	23.2
Angiotensin converting enzyme inhibitors/angiotensin receptor blocker user, %	30.3
Statin user, %	49.7
Bacteria species	
*Firmicutes*, copies ∗ 10^9^/g	5.90 (3.11, 9.91)
*Bacteroidetes*, copies ∗ 10^9^/g	9.73 (4.72, 17.19)
*Firmicutes/Bacteroidetes*	0.60 (0.30, 1.19)
*C. leptum* group, copies ∗ 10^8^/g	8.01 (2.96, 14.89)
*Bacteroides*, copies ∗ 10^9^/g	2.02 (1.06, 3.83)
*Bi**fidobacterium*, copies ∗ 10^6^/g	2.74 (0.26, 15.60)
*A. muciniphila*, copies ∗ 10^5^/g	0.14 (0.05, 53.35)
*E. coli*, copies ∗ 10^8^/g	1.17 (0.32, 5.59)
*F. prausnitzii*, copies ∗ 10^7^/g	12.41 (2.56, 31.71)
Laboratory parameters	
Glycated hemoglobin, %	7.0 (6.5,7.9)
Hemoglobin, g/dL	11.0 ± 8.6
Albumin, g/dL	4.6 (4.3, 4.8)
Uric acid, mg/dL	5.9 ± 1.4
Cholesterol, mg/dL	168 ± 39
Triglyceride, mg/dL	121 (81,170)
High-density lipoprotein, mg/dL	44 ± 12
Low-density lipoprotein, mg/dL	89 (72,111)
Creatinine, mL/min/1.73 m	1.0 ± 0.5
Urine albumin/creatinine ratio, mg/g	16.3 (6.8, 57.8)

Data are expressed as the number (percentage) for the categorical variables and the mean ± SD or median (25th, 75th percentile) for the continuous variables, as appropriate.

**Table 2 nutrients-13-02679-t002:** The echocardiographic parameters of the study patients.

Echocardiographic Parameters	Entire Cohort N = 155
Aortic root diameter, cm	3.3 ± 0.5
Left atrium diameter, cm	3.7 ± 0.7
Left atrium diameter/Aortic root diameter	1.1 ± 0.3
left ventricular mass index, g/m^2^	95.8 ± 38.4
left ventricular hypertrophy, *n* (%)	29(18.7)
stroke volume, ml	82.6 ± 24.4
left ventricular ejection fraction, %	70.6 ± 10.1
Left ventricular fraction shortening, %	40.9 ± 8.1
E/A ratio	0.8 ± 8.1
E/A ratio <1, *n* (%)	115(79.9)
E/E’ ratio	9.2 ± 2.8
E/E’ ≥ 15, *n* (%)	4(2.8)

Data are expressed as the number (percentage) for the categorical variables and mean ± SD or median (25th, 75th percentile) for the continuous variables, as appropriate. Abbreviations: E/A, peak early transmitral filling wave velocity/peak late transmitral filling wave velocity; E/E’, early mitral inflow velocity/mitral annular early diastolic velocity.

## Data Availability

The data presented in this study are available on request from the corresponding author. The data are not publicly available due to privacy.

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
