# Peer review of "Gut Microbiota and Subclinical Cardiovascular Disease in Patients with Type 2 Diabetes Mellitus"

_nutrients, 2021, doi:10.3390/nu13082679_

Round 1
Reviewer 1 Report
The authors investigated the relationship between gut microbiota content and progression of type 2 diabetes (T2D) and cardiovascular disease (CVD). They measured gut microbiota contents in feces of T2D patients and analyzed the relationship with cardiac structure and function in T2D patients, and found that the different contents of gut microbiota are associated with cardiovascular abnormalities. They concluded that the changes of gut microbiota contents may be associated with the development of subclinical CVD in T2D patients. The data are sound. However, there are concerns that should be addressed. The description of Tables 3-5 should be improved. Moreover, I think it is difficult to predict CVD from changes in gut microbacteria content.
- I cannot understand the meaning of medium bacterial concentrations shown in Table 1. What did the authors would like to express using these data?
- Although the data of Tables 3-5 are important, they are very difficult to see and understand. They had better be shown by graphs. It is necessary better ways to make the reader understand.
- The specificities of amplification of each microbacteria should be shown as Supplemental data.
- Line 150: 1 μl of sample DNA was used. However, the amount of DNA used in qPCR should be indicated, but not volume.
- The authors concluded that the imbalance of gut microbiota might be a potential biomarker of the development of subclinical CVD in patients with T2D. However, many factors affect to the ratio of microbiota. How do the authors judge the development of subclinical CVD from the imbalance of microbiota? This information can be used as biomarker? It is questionable.
Author Response
Thanks for your precise comments. We do our best to revise this manuscript including the description of results and content depending on your suggestion.
Question 1: I cannot understand the meaning of medium bacterial concentrations shown in Table 1. What did the authors would like to express using these data?
Answer 1: Because the bacterial concentrations are non‑normal distribution, we use medium (25th percentile and 75th percentile) to express. The presentation is depending on other studies including reference NO. 30, 31 and 32.
- Guo, X.; Xia, X.; Tang, R.; Wang, K. Real-time PCR quantification of the predominant bacterial divisions in the distal gut of Meishan and Landrace pigs. Anaerobe 2008, 14, 224-228.
- Penders, J.; Thijs, C.; Vink, C.; Stelma, F.F.; Snijders, B.; Kummeling, I.; van den Brandt, P.A.; Stobberingh, E.E. Factors influencing the composition of the intestinal microbiota in early infancy. Pediatrics 2006, 118, 511-521.
- Karlsson, C.L.; Onnerfalt, J.; Xu, J.; Molin, G.; Ahrne, S.; Thorngren-Jerneck, K. The microbiota of the gut in preschool children with normal and excessive body weight. Obesity (Silver Spring) 2012, 20, 2257-2261.
Question 2: Although the data of Tables 3-5 are important, they are very difficult to see and understand. They had better be shown by graphs. It is necessary better ways
to make the reader understand.
Answer 2: Thanks for your precise comment. We have revised our results shown by forest plot graphs. Please see Figure 2,3,4.
Question 3: The specificities of amplification of each microbacteria should be shown as Supplemental data.
Answer 3: Thanks for your suggestion. We used bacteria primers according to previous studies. Please see Supplemental Table S1
Question 4: Line 150: 1 μl of sample DNA was used. However, the amount of DNA used in qPCR should be indicated, but not volume.
Answer 4: Thank you very much for your comment. The qPCR was performed as previously described (the newly added references NO. 30,31 and 32). First, the fecal samples were weighted to be 50-100 mg. The DNAs were extracted by Stool DNA Extraction kit and were eluted in an equal volume, 35 μl. Finally, 1 μl, an equal volume of the extracted DNAs from each samples were used in qPCR. Therefore, the quantity of bacteria was expressed as the 16S rRNA gene copy numbers/g weight of feces. We add the description “The purified DNA was eluted in a volume of 35 μl” in the Section 2.3 to fill the vacancy in the previous version.
Question 5: The authors concluded that the imbalance of gut microbiota might be a potential biomarker of the development of subclinical CVD in patients with T2D. However, many factors affect to the ratio of microbiota. How do the authors judge the development of subclinical CVD from the imbalance of microbiota? This information can be used as biomarker? It is questionable.
Answer 5: Thanks for your important suggestion. The correlation between gut microbiota and subclinical CVD is complicated and might be affected by many factors, such as diet habit and medication. This study has collected the information of diet habit and medication. All of the patients were enrolled in an education program on diabetes at our hospital, and they followed the guidelines for a diabetic diet to minimalize the effect of diet on gut microbiota. Accumulating evidence has been indicated that the impact of β-blocker, ACRI/ARB, statin on subclinical CVD. Hence, we adjusted the use of β-blocker, ACRI/ARB, statin in multivariate regression model of correlation between gut microbiota and subclinical CVD, and found the imbalance composition of gut microbiota was associated with abnormal cardiac structure and function. We have added the description in our in our discussion. Please see Page 12, Discussion, paragraph 2.

Reviewer 2 Report
The authors et al. have investigated the subclinical CVD in T2D and the role of gut microbiota in patients. To do so, they have select 155 patients and collected fecal, echo, and demographics data. They found that the low abundances of phyla Bacteroidetes and Firmicutes have correlated with diastolic dysfunction and a significant risk factor in CVD. While the paper is of great interest, there are few comments to be addressed before the publication.
- qPCR quantification should be considered as equivalent to NGS technologies, it would have been a wonderful opportunity to perform NGS from these valuable patient fecal samples to show alpha, beta and overall compositional changes in the gut microbiota.
- The overall generalizability of the results got hampered due to low sample numbers and relatively less diversity and use of single site/hospital for recruitment.
- Methods: Patients who have received antibiotics one month prior were excluded, but according to microbiome research, it can take at least 2-3 months to recover from antibiotic-induced dysbiosis.
- Moreover, concomitant medication may have a huge role to play in microbiome changes, needs to be carefully discussed here.
- Where the patient enrolled include female sex, as the table 1 and methods needs to be updated.
- The diabetic diet, fiber% may influence the microbiome. Particularly, Akkermansia muciniphila, has a huge role to play in T2D.
- Also, I suggest to update the table which will enable to differentiate the changes between the conditions.
Author Response
Question 1: qPCR quantification should be considered as equivalent to NGS technologies, it would have been a wonderful opportunity to perform NGS from these valuable patient fecal samples to show alpha, beta and overall compositional changes in the gut microbiota.
Answer 1: Thanks for your valuable suggestion. Lacking next generation sequencing (NGS) analysis of gut microbiota is the limitation of this study. NGS analysis might help us to realize all picture of the relative composition of gut microbiota. The aim of our study was to measure absolute concentration of target bacteria related to T2D patients, and further analyze the relationship between target bacteria and subclinical CVD. Further study is necessary to perform NGS analysis to show alpha, beta and overall compositional changes of the gut microbiota in T2D patients to provide more detailed information about the correlations between echocardiographic changes and gut microbiota. We have revised the description of limitation. Please see Page 12, last paragraph in “Discussion”
.
Question 2: The overall generalizability of the results got hampered due to low sample numbers and relatively less diversity and use of single site/hospital for recruitment.
Answer 2: Thanks for your valuable comment. This is one of our limitations. The relatively small number of T2D subjects and single hospital for recruitment may have underestimated the association between the gut microbiota and subclinical CVD. We have revised the description of limitation. Please see Page 12, last paragraph in “Discussion”
Question 3: Methods: Patients who have received antibiotics one month prior were excluded, but according to microbiome research, it can take at least 2-3 months to recover from antibiotic-induced dysbiosis.
Answer 3: Thank you very much for your comment. There are no strict criteria for the time of recovery from antibiotic-induced dysbiosis. Wu et al. (Curr Microbiol. 61:69–78, 2010) excluded the patients who had a history of gastrointestinal diseases nor received antibiotics, probiotics and prebiotics, within 30 days. Tims et al. (ISME J. 7: 707–717, 2013) excluded the subjects who used medication that may affect the GI microbiota, prebiotics or probiotics within 1 month before sampling. Hamasaki-Matos et al. (BMC Res Notes. 14:238, 2021) describe the exclusion criteria including use of antibiotics in the previous 4 weeks. Therefore, we excluded the patients who have received antibiotics one month in the current study. We add these references (NO. 27,28 and 29) in Materials and Methods 2.1.
- Wu, X.; Ma, C.; Han, L.; Nawaz, M.; Gao, F.; Zhang, X.; Yu, P.; Zhao, C.; Li, L.; Zhou, A.; Wang, J.; Moore, J.E.; Millar, B.C.; Xu, J. Molecular characterisation of the faecal microbiota in patients with type II diabetes. Curr Microbiol 2010, 61, 69-78.
- Tims, S.; Derom, C.; Jonkers, D.M.; Vlietinck, R.; Saris, W.H.; Kleerebezem, M.; de Vos, W.M.; Zoetendal, E.G; Microbiota conservation and BMI signatures in adult monozygotic twins. ISME J. 2013, 7, 707-17.
- Hamasaki-Matos, A.J.; Cóndor-Marín, K.M.; Aquino-Ortega, R.; Carrillo-Ng, H.; Ugarte-Gil, C.; Silva-Caso, W.; Aguilar-Luis, M.A.; Del Valle-Mendoza, J. Characterization of the gut microbiota in diabetes mellitus II patients with adequate and inadequate metabolic control. BMC Res Notes 2021, 14, 238).
Question 4: Moreover, concomitant medication may have a huge role to play in microbiome changes, needs to be carefully discussed here.
Answer 4: Thanks for your valuable suggestion. The correlation between gut microbiota and subclinical CVD is complicated and might be affected by many factors, such as diet habit and medication. For example, previous studies suggested that metformin may be related to higher abundance of A muciniphila and several butyrate-producing bacteria to maintain glycemic status in T2D patients. Kim et al. reported that statin therapy may increase the abundance of the genera Bacteroides. This study has collected the information of diet habit and medication. All of the patients were enrolled in an education program on diabetes at our hospital, and they followed the guidelines for a diabetic diet to minimalize the effect of diet on gut microbiota. We have analyzed the correlation between medication and target bactereia in these patients, and did not find the significant effect of medication on target bactereia concentration. In addition, accumulating evidence has been indicated that the impact of β-blocker, ACRI/ARB, statin on subclinical CVD. Hence, we adjusted the use of β-blocker, ACRI/ARB, statin in multivariate regression model of correlation between gut microbiota and subclinical CVD, and found the imbalance composition of gut microbiota was associated with abnormal cardiac structure and function.We have added the description in our in our discussion. Please see Page 12, Discussion, paragraph 2.
Question 5: Where the patient enrolled include female sex, as the table 1 and methods needs to be updated.
Answer 5: We have updated Table 1 and methods.
Question 6: The diabetic diet, fiber% may influence the microbiome. Particularly, Akkermansia muciniphila, has a huge role to play in T2D.
Answer 6: Thanks for your valuable suggestion. The correlation between gut microbiota and subclinical CVD is complicated and might be affected by many factors, such as diet habit and medication. This study has collected the information of diet habit and medication. All of the patients were enrolled in an education program on diabetes at our hospital, and they followed the guidelines for a diabetic diet to minimalize the effect of diet on gut microbiota. In addition, we didn’t found the relationship between A muciniphila and subclinical CVD in T2D patients. We have added the description in our in our discussion. Please see Page 12, Discussion, paragraph 2.
Question 7: Also, I suggest to update the table which will enable to differentiate the changes between the conditions.
Answer 7: Thanks for your valuable comment. We have revised our results shown by graphs. Please see Figure 2-4.

Reviewer 3 Report
In the manuscript ID- nutrients-1284705 titled “Gut microbiota and subclinical cardiovascular disease in patients with type 2 diabetes mellitus” by Hui-Ju Tsai and colleagues. The authors have reported that phyla Bacteroidetes and Firmicutes, and genera Bacteroides were positively correlated with left ventricular ejection fraction. High levels of phylum Firmicutes and genera Bacteroides were associated with a reduced risk of left ventricular hypertrophy. High levels of both phylum Bacteroidetes and genera Bacteroides were negatively associated with diastolic dysfunction. A high phylum Firmicutes/Bacteroidetes (F/B) ratio and low level of genera Bacteroides were correlated with increased left atrial diameter. Phyla Firmicutes and Bacteroidetes, F/B ratio, and genera Bacteroides were associated with variations in cardiac structure and systolic and diastolic dysfunction in T2D patients. I have few concerns regarding the present manuscript.
-The introduction section requires more information about the relationship between the gut microbiota and T2D and CVD, key species, or key and important molecules more than TMAO.
-The present study is collected on any websites, e.g. clinicaltrials.gov or someone similar?
-Who the authors have calculated the sample size for the present study?
-I read with interest the material and methods section, to evaluate which technique was used for microbial analysis, in the author’s opinion, this technique is better than 16S novel techniques, Illumina or even Nanopore?
-About the statistical analysis, more than correlations or even principal component analysis could be better for the authors in the present manuscript.
-My main concern about the present manuscript is the feasibility of microbial detection, and the general view of the data compared with the novel 16S techniques aforementioned.
Author Response
Question 1: The introduction section requires more information about the relationship between the gut microbiota and T2D and CVD, key species, or key and important molecules more than TMAO.
Answer 1: Thanks for your suggestion. We have revised the introduction. Please see Page 2, Paragraph 3
Question 2: The present study is collected on any websites, e.g. clinicaltrials.gov or someone similar?
Answer 2: The present study is not collected on any websites.
Question 3: Who the authors have calculated the sample size for the present study?
Answer 3: Yi-Chun Tsai and Wei-Wen Hung calculated the sample size for the present study.
Question 4: I read with interest the material and methods section, to evaluate which technique was used for microbial analysis, in the author’s opinion, this technique is better than 16S novel techniques, Illumina or even Nanopore?
Answer 4: NGS and qPCR both offer highly sensitive and reliable variant detection of the composition of gut microbiota. The qPCR only detects known sequences. In contrast, NGS is a hypothesis-free approach that does not require prior knowledge of sequence information. NGS provides higher discovery power to detect novel genes and higher sensitivity to quantify rare variants and transcripts. The qPCR is effective for low target numbers, and the workflow can be cumbersome for multiple targets. NGS is preferable for studies with many targets or samples. The choice between NGS vs. qPCR depends on several factors, including the number of samples, the total amount of sequence in the target regions, budgetary considerations, and study goals. The qPCR is typically a good choice when the number of target regions is low (≤ 20 targets) and when the study aims are limited to screening or identification of known variants. The aim of our study was to measure absolute concentration of target bacteria related to T2D patients, and further analyze the relationship between target bacteria and subclinical CVD. Further study is necessary to perform NGS analysis to show alpha, beta and overall compositional changes of the gut microbiota in T2D patients.
Question 5: About the statistical analysis, more than correlations or even principal component analysis could be better for the authors in the present manuscript.
Answer 5: Thanks for your suggestion. We have added principal component analysis as Figure 1. Figure 1 showed phylum Firmicutes was negatively associated with LVMI, and phylum Bacteroidetes, genera Bacteroides and F/B ration was negatively related to LA diameter using PCA.
Question 6: My main concern about the present manuscript is the feasibility of microbial detection, and the general view of the data compared with the novel 16S techniques aforementioned.
Answer 6: Thanks for your suggestion. The aim of our study is to evaluate the relationship between targeted gut microbiota and subclinical CVD in T2D patients. The qPCR is typically a good choice when the number of target regions is low (≤ 20 targets) and when the study aims are limited to screening or identification of known variants. The aim of our study was to measure absolute concentration of target bacteria related to T2D patients, and further analyze the relationship between target bacteria and subclinical CVD. Therefore, we use qPCR in this study. Further studies are needed with next generation sequencing to analyze the whole profile of gut microbiota to provide more detailed information about the correlations between echocardiographic changes and gut microbiota. Please see Page 12, last paragraph.

Round 2
Reviewer 1 Report
Manuscript was improved. I have no more comment.
Author Response
We appreciate for Editor and Reviewers’ precise comments to make the manuscript suitable for publication in the eminent journal.
Reviewer 3 Report
Thank you to the authors for addressed the majority of my concerns about the previous manuscript, the most important issue is the microbial analysis (qPCR vs. NGS
Author Response
Thanks for your valuable suggestion. We have revised the description about qPCR vs. NGS in the discussion. Please see Page 12, last paragraph
Finally, this study only measured eight targeted gut bacteria using real-time qPCR. There may be other bacteria related to echocardiographic changes that we did not examine. Lacking next generation sequencing (NGS) analysis of gut microbiota is one of our limitations. NGS is a hypothesis-free method that does not require prior knowledge of sequence information and provides higher discovery power to detect novel genes and higher sensitivity to quantify rare variants and transcripts. NGS analysis might help us to realize all picture of the relative composition of gut microbiota. Because the aim of our study was to analyze the relationship between target bacteria and subclinical CVD, qPCR is typically a good choice when the number of target regions is low (≤ 20 targets) and when the study aims are limited to identify the known variants. Nevertheless, further study is also necessary to perform NGS analysis to show alpha, beta and overall compositional changes of the gut microbiota in T2D patients to provide more detailed information about the correlations between echocardiographic changes and gut microbiota.